# Design of a Hybrid Energy System with Energy Storage for Standalone DC Microgrid Application

**Mwaka I. Juma [1,2,\*], Bakari M. M. Mwinyiwiwa [1], Consalva J. Msigwa [2] and Aviti T. Mushi [1]**

1. Electrical Engineering Department, University of Dar es Salaam, Dar es Salaam P.O. Box 35131, Tanzania; bakari1mwinyiwiwa@gmail.com (B.M.M.M.); aviti.thadei@udsm.ac.tz (A.T.M.)
2. Electrical Engineering Department, Dar es Salaam Insititue of Technology, Dar es Salaam P.O. Box 2958, Tanzania; msigwaj34@gmail.com
* Correspondence: jumamwaka@gmail.com

**Abstract:** This paper presents microgrid-distributed energy resources (DERs) for a rural standalone system. It is made up of a solar photovoltaic (solar PV) system, battery energy storage system (BESS), and a wind turbine coupled to a permanent magnet synchronous generator (WT-PMSG). The DERs are controlled by maximum power point tracking (MPPT)-based proportional integral (PI) controllers for both maximum power tracking and error feedback compensation. The MPPT uses the perturb and observe (P&O) algorithm for tracking the maximum power point of the DERs. The PI gains are tuned using the Ziegler–Nichols method. The developed system was built and simulated in MATLAB/Simulink under two conditions—constant load, and step-load changes. The controllers enabled the BESS to charge even during conditions of varying load and other environmental factors such as change of irradiance and wind speed. The reference was tracked extremely well by the output voltage of the DC microgrid. This is useful research for electrifying the rural islanded areas which are too far from the grid.

**Keywords:** solar photovoltaic (PV); wind turbine coupled to permanent magnet synchronous generator (WT-PMSG); battery energy storage system (BESS); maximum power point tracking (MPPT); DC/DC converters

## 1. Introduction

Recent research has shown that in Tanzania, the access to electricity is limited to 35.6% of the total 56.32 million population as of 2018 [1]. One location, a village called Luxmanda, is not connected to the grid, but survives on the available DC microgrid which provides power to some few loads [2]. This problem of limited access to electricity can be reduced not only by grid expansions, but rather by utilizing distributed renewable energy sources (RES). These systems, such as solar home systems (SHS) and micro- and mini-solar plants, are increasingly being used as sources of electric energy in rural areas worldwide and in Tanzania. They are designed for use at the demand of small households, usually in power ranging by few kilowatts, thereby causing limitations for enterprise and other potentially larger users of electricity within rural areas. However, if these hybrid RES are deployed in villages or remote locations, they result to least net present cost and reduced emission of carbon dioxide [3,4] by paying extra attention to their optimal sizing design as was carried out in Palestine [5]. These microgrids can be highly efficient in delivering energy to local loads [6].

Solar photovoltaic (PV) plants and wind energy need big power storage (such as batteries) to provide voltage regulation, and reduce the effects of the energy source intermittency, which adds to the cost of the installation. For example, in India [6], a solar PV plant was able to supply 55.1% of the required electricity. This, however, still leaves the locale dependent on the grid. To reduce the cost of battery, it has been proposed to integrate several RES to form a hybrid power system [7]. Therefore, there exists a need to manage the

flow of energy on these hybrid RES to ensure the reliability and availability of power supply to meet load demand. However, control of these hybrid RES systems is usually a difficult task [8]. One highly complex task is to control these hybrid RES-distributed energy sources (DERs) in a micro grid to maintain voltage of the micro grid within an acceptable range of $\pm 5\%$ of the DC bus voltage [9]. The DC microgrid at Luxmanda village suffers from voltage variations. This paper will review briefly previous proposed control architectures and point out their limitations as pertaining to the DC microgrid of Luxmanda village.

The model-based design approach for stability and controller tuning was proposed by Petersen et al. [10]. The traditionally employed active–reactive (PQ) droop control has been applied in power control of a microgrid, which has success in grid-connected mode. However, the PQ droop control fails miserably when the microgrid goes into islanded mode due to a poor reactive and resistive (X/R) ratio [11,12]. Another researcher [13] proposed to limit the stochastically varying DERs output by interfacing them to the DC grid using a SEPIC converter. The method works well to eliminate the effects of SEPIC voltage input (from the DERs) variation and step-load changes. Further, artificial neural network (ANN) and fuzzy logic controllers (they can be summed up as artificial intelligence–AI) have been employed to satisfy load requirements from a microgrid [14,15]. However, the large number of connected DERs, which sometimes impose those AI controllers to fulfil conflicting requirements, is fraught with limited communication. The AI-based controllers are computationally intensive [15]. Others [16] reviewed the decentralized controller framework for the microgrids, with the caveat that there is still the problem of large-scale interconnections of the microgrids. This has been partially solved through microgrid central controller (MGCC) installed either on the low voltage side or low voltage substation [17]. This architecture is complex, expensive, and is not feasible for implementation in DERs located in rural areas of many developing nations. Proof of concept control of stand-alone microgrids was implemented by Petersen et al., [10] both by simulation and on hardware in the loop approach. However, this method involves too many steps of execution.

There have been active studies on stability of microgrids connected to the main grid [18–20] with some promise of good performance. The model for the islanded microgrid is developed by integrating all the inverter dynamics using a state-space model for the load currents [21]. However, these studies are not focused on voltage control of DC microgrids for rural areas. Chauhan et al. [22] presented a control scheme for a wind generator (WG) coupled with a battery energy storage. This control ensured synchronization of the WG and disconnection during velocity variations.

Voltage regulation on the DC bus of a microgrid has been accomplished extensively in the surveyed literature [2,11,23–26]., However, voltage dip and swell still occurs whenever the inputs, control, and outputs (loads connected to the DC bus) change abruptly. Tan et al. [27] proposed an MPPT that is based on the fluctuations of the output voltage, power, and duty cycle. High-performance photovoltaic constant power generation control and MPPT was proposed, and worked excellently in change of irradiance and temperature [28]. However, the focus of these papers was not on the DC bus voltage control, and also did not consider hybrid RES. Therefore, this paper proposes a control system for the RES to maintain the DC bus voltage at 750 V, irrespective of the varying solar irradiance, DC microgrid load step change, and wind speed changes. The main contributions of the paper are as follows. The detailed design of the standalone DC microgrid, the detailed design of the controller for the DC bus voltage which is the MPPT-based PI control algorithm incorporating DC bus voltage regulation. Those PI gains were tuned by the Ziegler–Nichols method. Another contribution is implementing the HRES and its controller in MATLAB/Simulink for simulation purposes and providing preliminary results for the Luxmanda village DC microgrid voltage control.

This paper is organized as follows. Section 2 presents the materials and methods in detail. Section 3 presents the developed controller algorithms for solar PV, and DC bus voltage regulation. Section 4 presents the simulation and results for cases of constant/increasing loading of the DC bus. Section 5 concludes the paper and sets out the future work.

## 2. Materials and Methods

This paper considers the following energy resources constituting the microgrid: DERs for a rural standalone system at Luxmanda village solar PV plant rated 31.5 kW; wind turbine equipped with permanent synchronous generator (WT-PMSG) rated 6 kW; and battery storage rated 248 Ah (6.4 kW), respectively, as shown in Figure 1. The solar PV plant gives out intermittent power from the solar irradiation at a DC voltage of about 547 V, thus the DC/DC boost converter steps up this voltage to a steady output of 750 V and feeds it to the DC bus. The WT-PMSG gives intermittent output AC voltage of 500 V, which is rectified by the AC/DC rectifier to DC voltage of 477 V, and then this voltage is stepped up to 750 V through the boost DC/DC converter. Lastly, but not the least is the battery energy storage system (BESS) at 240 V DC. The battery is charged through the bidirectional DC/DC converter (BDC) and discharges through the same. Therefore, this section will discuss the modelling of the solar PV array, WT-PMSG, and the DC/DC boost converter.

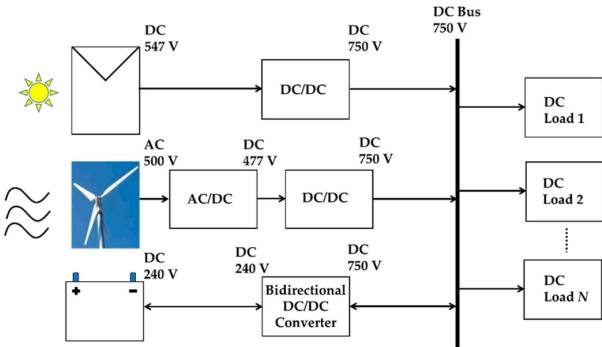

**Figure 1.** Microgrid DERs for a rural standalone system.

### 2.1. Solar PV Array Modelling

Solar PV array is made up of series ($N_{SS}$) connected modules, and a string of these modules are together strung in a parallel ($N_{PP}$) connection. Therefore, using the diode model of a solar cell [29], the total current $I_{PV,Tot}$ that comes out of the solar PV array is presented by [30] in (1)–(4). The solar module current is $I_{PV}$, $R_S$ and $R_P$ are solar cell series and shunt resistances respectively. The adopted diode model of a solar cell has dark saturation current $I_0$, harvested solar cell current $I_{PV}$, diode ideality factor *a*, diode thermal voltage $V_T$. The Boltzmann constant is $k = 1.380 \times 10^{-23}$ J/K, $T$ is the temperature of the cell, and $q = 1.6022 \times 10^{-19}$ C is the elementary charge. The voltage ratio $\kappa$ and current ratio $\chi_I$ are used to simplify Equations (1)–(3).

$$I_{PV,Tot} = N_{PP}(I_{PV} - I_0(e^\kappa - 1)) - \chi_I \tag{1}$$

$$\kappa = \frac{N_{PP}V_{PV,Tot} + N_{SS}R_S I_{PV,Tot}}{aN_{SS}N_{PP}V_T} \tag{2}$$

$$\chi_I = \frac{N_{PP}V_{PV,Tot} + N_{SS}R_S I_{PV,Tot}}{N_{SS}R_P} \tag{3}$$

$$V_T = \frac{kT}{q} \tag{4}$$

The $I_0$ is a function of the bandgap energy $E_g = 1.12$ eV at nominal temperature $T_n = 25$ °C, with a nominal saturation current $I_{0,n}$ as captured by (5)–(8). Solar cell nominal short circuit current is represented by $I_{sc,n}$, with nominal open circuit voltage represented by $V_{oc,n}$. For series $N_{SS}$-connected solar modules, the thermal voltage is $V_{T,n}$. Other placeholder variables defined by $\varpi$, and $v$ are used for convenience.

$$I_0 = I_{0,n}\left(\frac{T_n}{T}\right)^3 e^\varpi \tag{5}$$

$$\omega = \frac{9E_g}{ak}\left(\frac{1}{T_n} - \frac{1}{T}\right) \tag{6}$$

$$I_{0,n} = \frac{I_{sc,n}}{e^v - 1} \tag{7}$$

$$v = \frac{V_{oc,n}}{aV_{T,n}} \tag{8}$$

The $I_{PV}$ dependence on the solar irradiance $G$ is presented by (9), where nominal irradiance is $G_n$ = 1000 W/m$^2$, $K_i$ is the short circuit current/temperature coefficient, and $I_{PV,n}$ is the nominal current at nominal conditions.

$$I_{PV} = (I_{PV,n} + K_i(T - T_n))\frac{G}{G_n} \tag{9}$$

### 2.2. Wind Turbine Modeling

The wind power $P_w$ extracted from the variable wind speed $V_w$, with turbine blades cutting an area $A$ is (10) as presented by Haque et al., and Wu et al. [31,32]. The Betz limit $C_p$ (11) is a function of blade pitch angle $\beta$, and tip speed ratio $\lambda$ (12). Radius of the shaft that is coupled to the rotor of the generator is $R$, rotating at an angular speed $w$. Betz limit has a maximum value equal to 0.593 [33].

$$P_w = 0.5\rho AV_w^3 C_p(\beta, \lambda) \tag{10}$$

$$C_p = 0.5\left(\lambda - 0.022\beta^2 - 5.6\right)e^{-0.17\lambda} \tag{11}$$

$$\lambda = \frac{wR}{V_w} \tag{12}$$

Smaller wind turbines have fixed $\beta$; therefore, for this case, $C_p$ is a function of $\lambda$ alone. For every $V_w$, there is a different optimal $\lambda$. Thus, coupling a generator with variable speed drives enables maximum power extraction at different $V_w$.

Modelling of Permanent Magnet Synchronous Generator

The permanent magnet synchronous generator (PMSG) that is coupled to the wind turbine is modeled in direct-quadrature (d-q) synchronous frame [32,33], by the following expressions.

$$v_d = -R_s i_d - L_d\frac{di_d}{dt} + L_q i_q w \tag{13}$$

$$v_q = -R_s i_q - L_q\frac{di_q}{dt} + L_d i_d w + \phi_m w \tag{14}$$

Direct and quadrature axes' voltages and currents are: $v_d$, $v_q$, $i_d$, and $i_q$, respectively. Respective inductances are $L_d$ and $L_q$. The stator resistance is denoted by $R_s$, while $w$ is the electrical angular speed. The PMSG establishes a magnetic flux linkage $\phi_m$, thus the electromagnetic torque $T_e$ is computed by (15), where $p$ is the number of pole pairs of the PMSG. Since the PMSG has a cylindrical rotor, $L_d = L_q$, then (15) simplifies to (16).

$$T_e = 1.5p\left((L_d - L_q)i_d i_q + \phi_m i_q\right) \tag{15}$$

$$T_e = 1.5p\phi_m i_q \tag{16}$$

### 2.3. DC/DC Boost Converter

The unregulated DC voltage output from the solar PV array or the rectifier of the PMSG needs to be regulated to a higher DC voltage of the DC bus as shown in Figure 1. This is accomplished by the DC/DC boost converter, whose steady state continuous current conduction mode operation is represented by (17) and (18), as found in [34–36]. The steady

state duty cycle is denoted by $D \in (0,1)$. The unregulated input voltage/current to the boost converter is denoted by $V_{in}/I_{in}$ and the regulated output voltage/current is denoted by $V_{DC}/I_{DC}$. Therefore, the boost converter's inductance $L$ and capacitance $C$ are computed in (19) and (20), respectively, following the work of [37]. The DC load is modeled by resistor $R$, and the boost converter is switching at a frequency of $f_s$ = 20 kHz, with an allowed voltage ripple $\Delta V_{DC}$.

$$V_{DC} = \frac{V_{in}}{1-D} \tag{17}$$

$$I_{DC} = I_{in}(1-D) \tag{18}$$

$$L \geq \frac{D(1-D)^2 R}{2 f_s} \tag{19}$$

$$C \geq \frac{D}{\Delta V_{DC} f_s R} \tag{20}$$

### 3. Maximum Power Point Tracking-Based Proportional and Integral Controller

This section discusses the maximum power point tracking (MPPT) control method that is used with the proportional integral (PI) controller to undertake the DC bus voltage regulation. One must consider the DC/DC boost converter with a source (e.g., solar PV array), and the MPPT-based PI controller shown by Figure 2. With the solar panel as shown in Appendix A Table A1, there is the input voltage $V_{PV}$ smoothing capacitor $C_{PV}$, and inductor $L_{PV}$ through which the input current $I_{PV}$ flows. The converter's switch $Q_{PV}$ turns ON and OFF at the switching frequency $f_s$ so that the diode delivers a charging current to the output capacitor $C_{dc}$. At steady state, the boost converter delivers output current $I_{dc}$ and voltage $V_{dc}$ to the DC microgrid.

The MPPT-based PI controller processes the measurements of $V_{PV}$, $I_{PV}$, and $V_{dc}$ to produce the switching duty cycle $D[k] \in (0,1)$, where $k$ is the discrete time instant. Each controller will be explained separately in the following subsections.

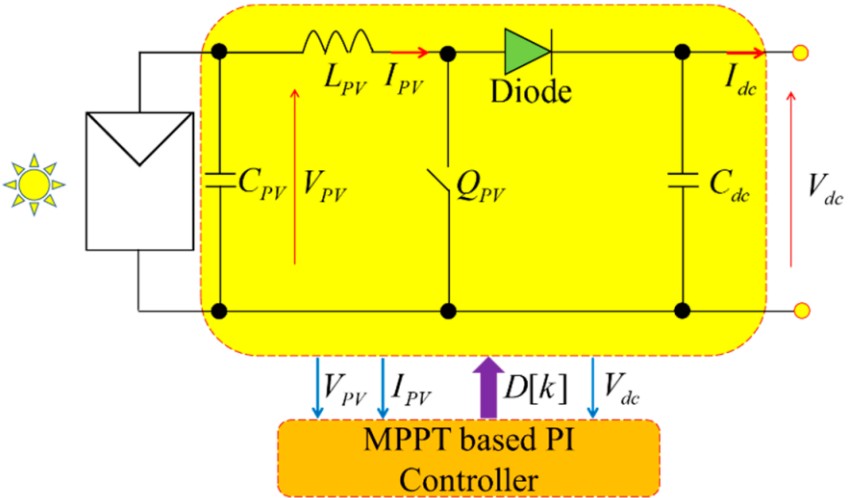

**Figure 2.** Equivalent circuit of boost DC/DC converter with MPPT-based PI control.

### 3.1. Maximum Power Point Tracking Controller

Researchers [23,38] have presented an MPPT technique using the Perturb and Observe (P&O) algorithm, which this paper adopts because of its simplicity to apply. In discrete time $k$ domain, the P&O algorithm tracks the maximum power by using the measurements of the input voltage $V[k]$ and current $I[k]$ of the source, then determining the power $P[k]$ which is used in adjusting the $D[k]$ accordingly to harvest maximum power. This duty

cycle $D[k]$ is computed from (21) by updating the previous sampling instant $k-1$ duty cycle $D[k-1]$ in (21) accordingly, as Figure 3 shows.

$$D[k] = D[k-1] \pm dD[k] \tag{21}$$

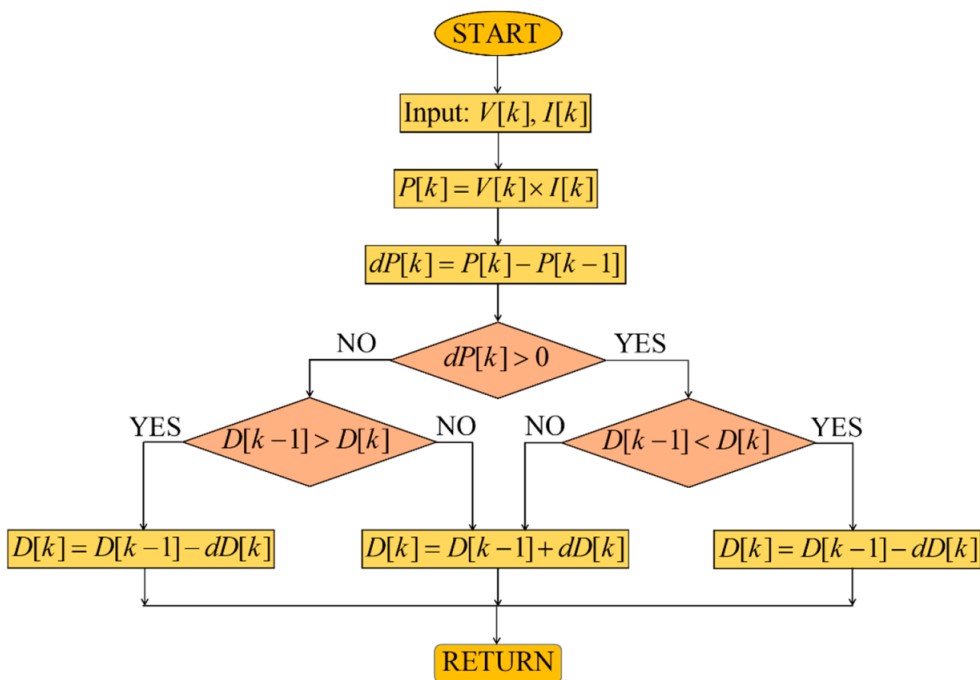

**Figure 3.** Flow chart of MPPT using P&O algorithm.

### 3.2. Proportional Integral Controller

The PI controller is used as a feedback compensation control algorithm in several industrial applications worldwide. For example, a PI controller was implemented to provide compensation for DC/DC boost converter [39] and a microgrid [40]. Given a system reference signal $r[k]$, output signal $y[k]$, where the output deviation from reference is $e[k] = r[k] - y[k]$, then it follows that the PI will provide control compensation $u[k]$ presented in (22) that is fed to the system to eliminate the deviation. The PI gains $k_{p1}$, $k_{p2}$, $k_{i1}$, and $k_{i2}$ are tuned using the Ziegler–Nichols method shown in Table 1.

$$u[k] = u[k-1] + k_P(e[k] - e[k-1]) + k_I e[k] \tag{22}$$

**Table 1.** Tuning the PI gains.

| Gain | Values |
|:---:|:---:|
| $k_{p1}$ | 0.5 |
| $k_{i1}$ | 0.6 |
| $k_{p2}$ | 0.5 |
| $k_{i2}$ | 0.001 |

### 3.3. DC Bus Voltage Regulation Design for Solar PV

The DC-DC Boost Converter regulates the DC bus voltage through MPPT control and PI control algorithms, discussed in the previous subsections. The MPPT control output is shown by (21), while the one from the PI is shown by (22). One should note that the reference $r[k]$ of (22) is replaced by $V_{dc,ref}[k]$ and the measured output $y[k]$ is replaced by the measured DC bus voltage $V_{dc}[k]$. Therefore, any discrepancy on the input or output is corrected by the MPPT and the PI controller in tandem, shown by Figure 4.

Thereafter, this paper proposes a PI which now performs additional corrective measures on the error generated from interaction between MPPT and the PI. To validate this new (proposed) control algorithm, the DC/DC boost converter with parameters found in Appendix A Table A2 [2,41] is simulated using MATLAB/Simulink. These parameters are adopted because they resemble those of the Luxmanda DC microgrid. The PI gains are tuned to those shown by Table 1.

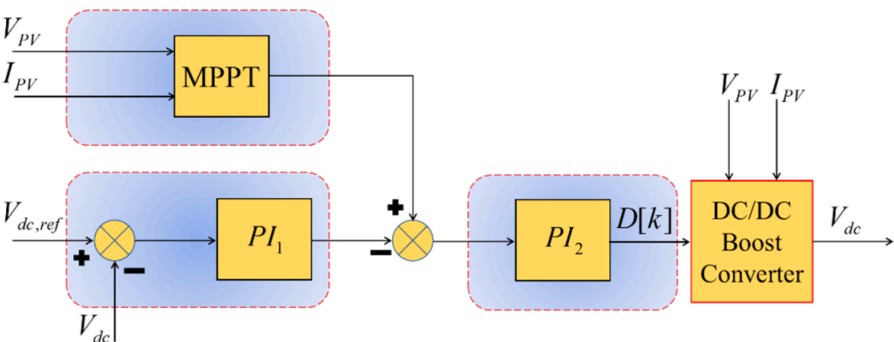

**Figure 4.** DC bus voltage regulation by MPPT-based PI control.

### 3.4. Solar PV with Battery Energy Storage System

Rajasekaran and Usha Rani [42] proposed a bidirectional DC/DC (BDC) converter to interface the battery to the DC microgrid. The interface is undertaken on the DC link with the boost DC/DC converter of the solar PV as shown in Figure 5. This converter is required to protect the battery from over and under charge through a suitable control algorithm. Furthermore, it has to have a wide current capacity to be capable of handling high current during low voltage operations. Similarly, this paper proposes to employ the BDC, shown by Figure 5 to maintain the DC microgrid voltage at 750 V. The battery voltage level is represented by $V_b$. The BDC inductor is $L_b$, which can pass the charge/discharge current $I_b$. The BDC switch is $Q_{b-b}$; it operates at a frequency of $f_s$. Depending on the design, it is the one responsible for the boost mode. The switch that is responsible for the buck mode is $Q_{c-d}$ operating at $f_s$. The battery charges in the buck mode, and discharges in the boost mode. Another research group [43] presented this converter as capable of adaptive power management and control, therefore it is suitable for this DC microgrid application.

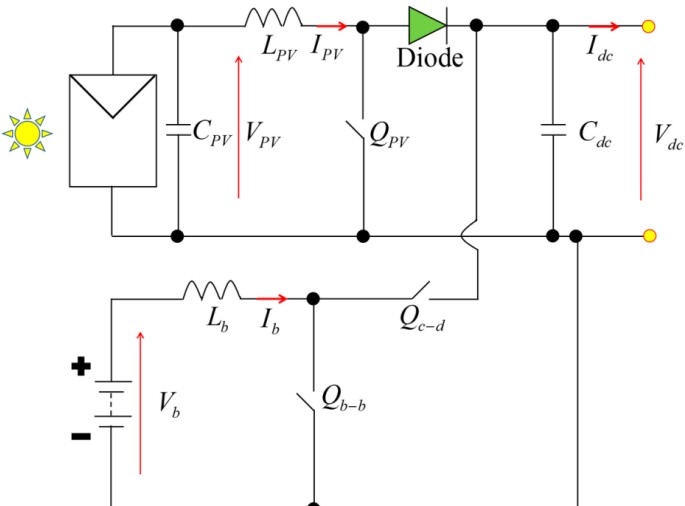

**Figure 5.** Solar PV with bidirectional buck–boost converter.

Denoting the duty cycle of $Q_{b-b}$ and $Q_{c-d}$ as $D_b$, one can design the BDC inductor by utilizing (23) and (24) as was conducted in [43], where the charging current ripple is denoted by $\Delta I_b$.

$$L_b = \frac{D_b(V_{dc} - V_b)}{f_s \Delta I_b} \tag{23}$$

$$D_b = \frac{V_b}{V_{dc}} \tag{24}$$

Control of Battery Energy Storage System

The charging and discharging conditions of the battery energy storage system (BESS) are tied to the state of charge (SOC), DC bus voltage, and net power ($P_{net}$) of the microgrid DERs for the rural standalone system. The $P_{net}$ is calculated as (25) [33], where the solar PV power is $P_{PV}$, wind power is $P_w$, and the load is $P_L$.

$$P_{net} = (P_V + P_W) - P_L \tag{25}$$

In case there is an excess of power from the sources ($P_{net} > 0$), the excess energy is used to charge the batteries. On the other hand, if it is not sufficient to power from the sources ($P_{net} < 0$), the battery energy is discharging as the proposed limit of SOC. This proposed battery control scheme is shown by Figure 6. If the battery SOC value is higher than 50%, the BESS discharges and supplies power to the load. The BESS charges when the SOC is less than 100%, while there is excess energy from the two sources. The battery charging and discharging control system utilizes two-loop PI structure, the outer loop with $PI_{b1}$ regulating the BESS voltage, and $PI_{b2}$ regulating its current. The boost and buck–boost converter parameters are found as per Appendix A Table A2. The capacity of the BESS is 12 V, 124 Ah, which are tabulated in Appendix A Table A3 [41], which are similar to the one in the Luxmanda DC microgrid. The calculated and tuned controller parameters of $PI_{b1}$ and $PI_{b2}$ are $k_{pb1} = 0.1$, $k_{ib1} = 1$, $k_{pb2} = 1$, and $k_{ib2} = 100$, respectively.

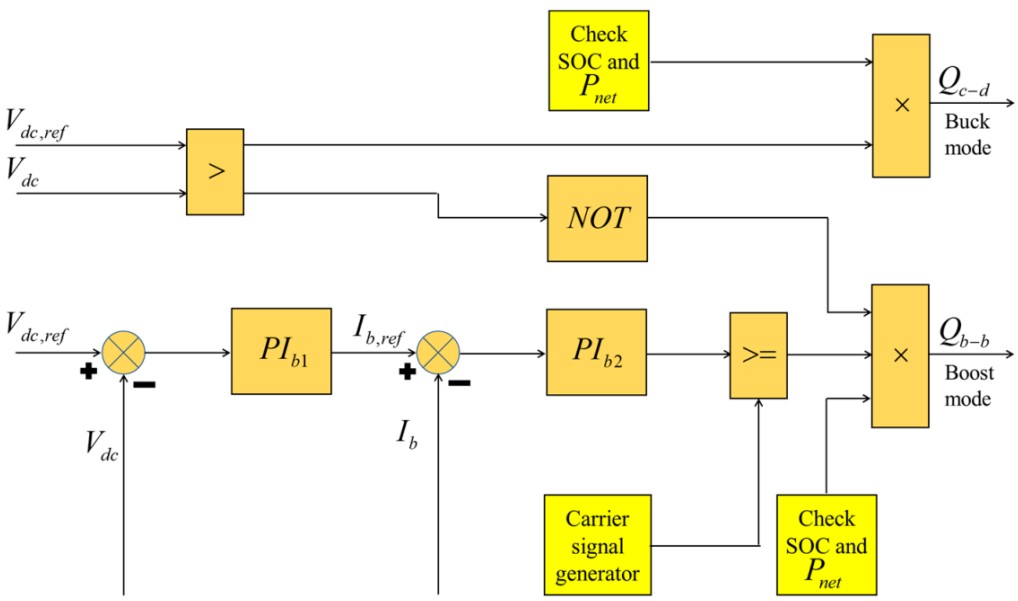

**Figure 6.** Control of BESS.

*3.5. DC Bus Voltage Regulation Design for Wind Generator*

The wind turbine energy conversion system consists of the wind turbine mechanically coupled to the permanent magnet synchronous generator (WT-PMSG). The parameters of PMSG are shown in Appendix A Table A4 [25]. The WT-PMSG is electrically connected to the rectifier, which is further coupled to the DC/DC boost converter. The MPPT is coupled to the rectifier and tracks the maximum power through rectifier voltage $V_d$ and

current $I_d$, with regard to the wind speeds $V_w$ of the WT-PMSG; it then calculates the duty cycle $D_1[k]$ as was conducted by [44], shown in Figure 7. The DC microgrid voltage is feedback-compensated by the PI controller $PI_w$ with gains tuned as follows: proportional gain, $k_{wp} = 0.00005$ and integral gain, $k_{wi} = 25$, which calculates the duty cycle $D_2[k]$. Therefore, the control system produces the duty cycle $D_w[k]$ by (25) as was conducted by [24,45,46].

$$D_w[k] = D_1[k] - D_2[k] \tag{26}$$

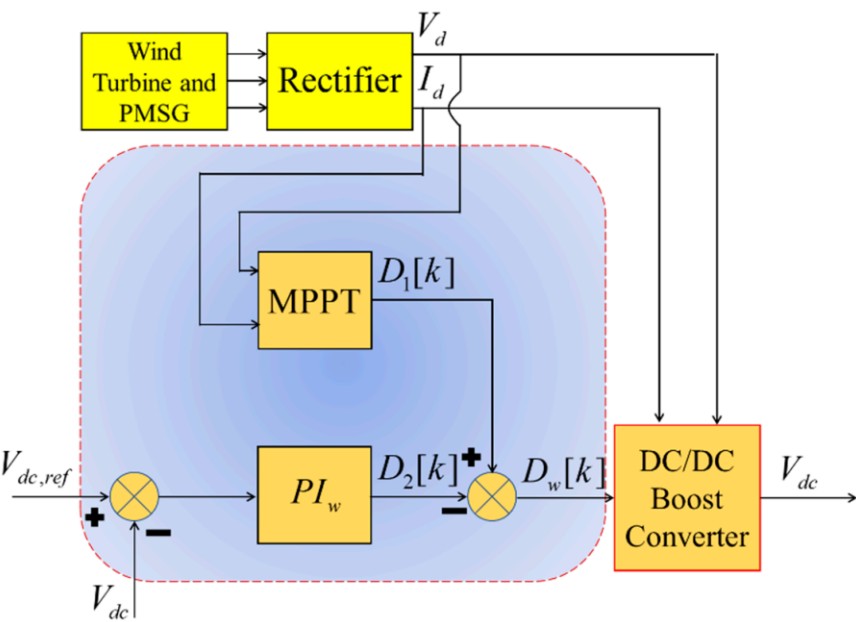

**Figure 7.** DC voltage regulator of WT-PMSG.

### 3.6. The Hybrid System of Solar-Wind with Battery Energy Storage System

The load demand is satisfied by the combination of solar PV, BESS, and WT-PMSG as shown in Figure 8. The WT-PMSG has the input-smoothing capacitor $C_d$, boost converter inductor $L_w$, and its switch $Q_w$.

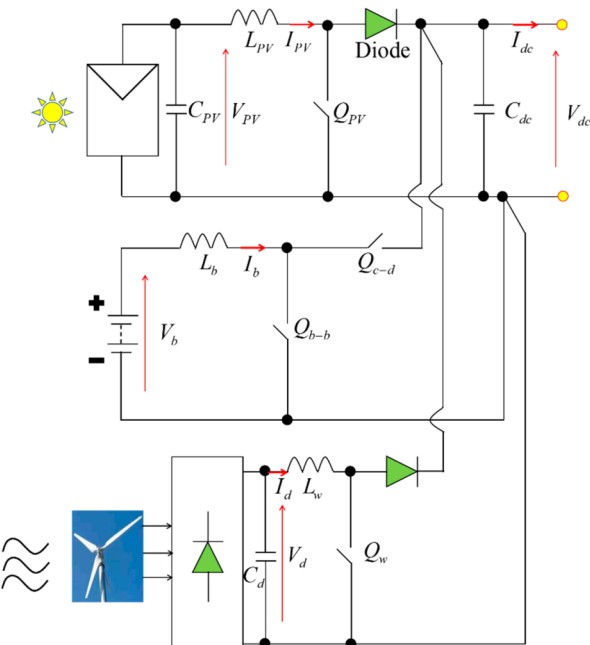

**Figure 8.** Circuit diagram of microgrid DERs for a rural standalone system.

## 4. Simulations and Results Discussions

The circuit diagram of a microgrid DERs for a rural standalone system was built in MATLAB/Simulink software, after design of all system parameters and controller gains. Two case studies were simulated—(1) constant load on the DC microgrid, and (2) step-load changes on the DC microgrid. Both scenarios considered constant temperature of 30 °C and solar irradiance between 800–1000 W/m$^2$, as shown in Figure 9. The wind speed was varied accordingly, with a mean of 12 m/s which translated to angular speed of 150–153 rad/s, as shown in Figure 9.

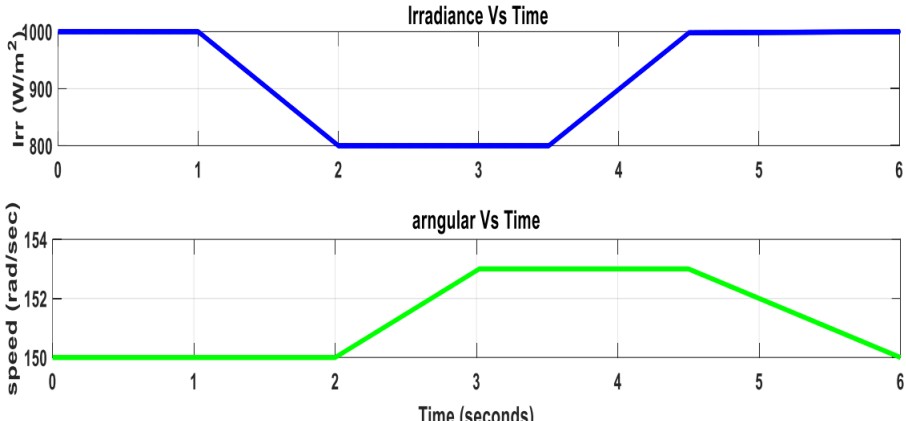

**Figure 9.** Variations of: solar irradiance Vs time (top curve) and angular speed vs time (bottom curve).

### 4.1. Case 1 Constant Loading of the DC Microgrid

Figure 9 shows that the solar irradiance is 1000 W/m$^2$ from 0–1 s, then it decreases linearly to 2 s. This irradiance is constant till 3.5 s, then it increases linearly to 1000 W/m$^2$ at 4.5 s. This irradiance is constant until the 6 s mark. The angular speed is 150 rad/s from the start till 2 s. The wind speed is increased linearly to 153 rad/s at the 3 s time. It stays constant for 1.5 s. Thereafter, it falls linearly to 150 rad/s at the 6 s mark. Figure 10 presents the results of the DC microgrid voltage at the constant loading, where it is observed reference ($V_{dc,ref} = 750$V) is tracked by the output voltage $V_{dc}$.

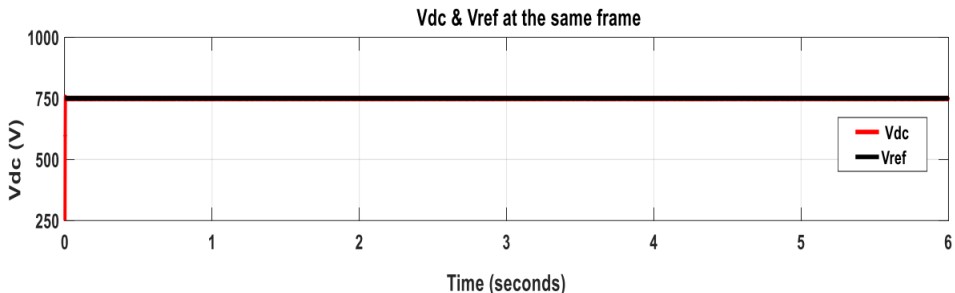

**Figure 10.** DC microgrid voltage at constant load.

### 4.2. Case 2 Step-Load Increase in the DC Microgrid

Figure 11 shows that the load current $I_{dc}$ on the microgrid starts from 20 A at 0 s. It stays the same for one second. Then, it increases step-wise to 30 A and stays the same until 2.5 s. Then, it jumps to 40 A at 2.5 s until 4.0 s. Thereafter, it jumps to 50 A at the 4.0 s mark. It stays this way until the 6.0 mark. For all these step-load variations, the DC microgrid voltage followed the reference, as shown in Figure 11.

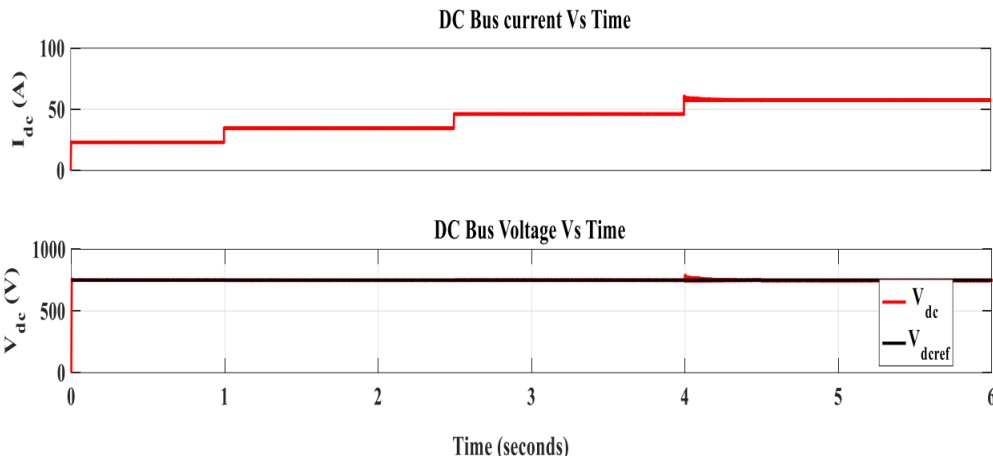

**Figure 11.** DC bus grid voltage when load step changes.

Figure 12 shows the battery voltage being maintained at 240 V while the battery current kept on increasing, thereby charging the battery, shown by the increment of SOC in the same Figure 12.

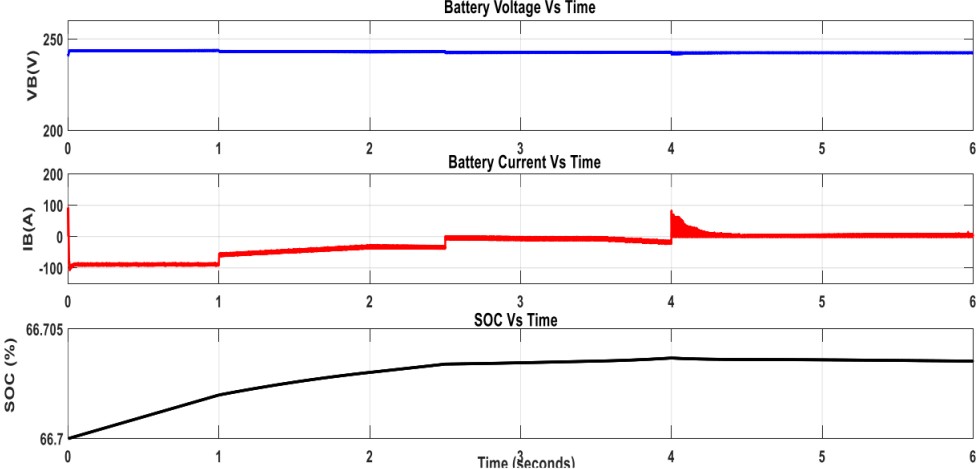

**Figure 12.** The BESS voltage, current and SOC during the load step change.

Figure 13 shows that the MPPT algorithms (for solar PV and WT-PMSG) could track the maximum power as desired. Figure 14 shows the solar PV MPPT achieving the MPP for irradiance between 800–1000 W/m$^2$. While all these happened, the output voltage could track the reference.

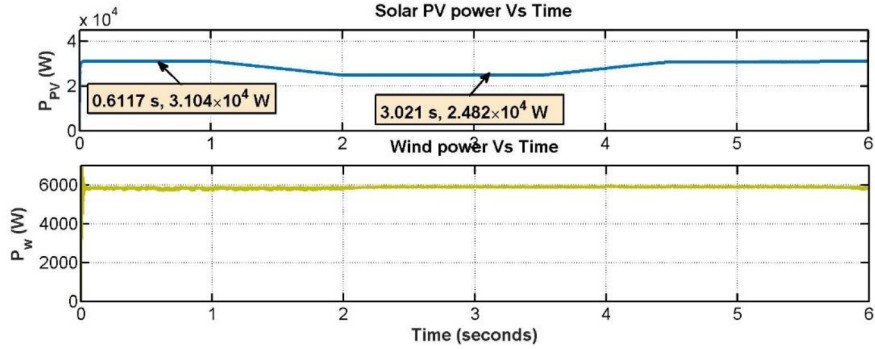

**Figure 13.** Power from solar PV and WT-PMSG achieved MPP through the MPPT algorithms.

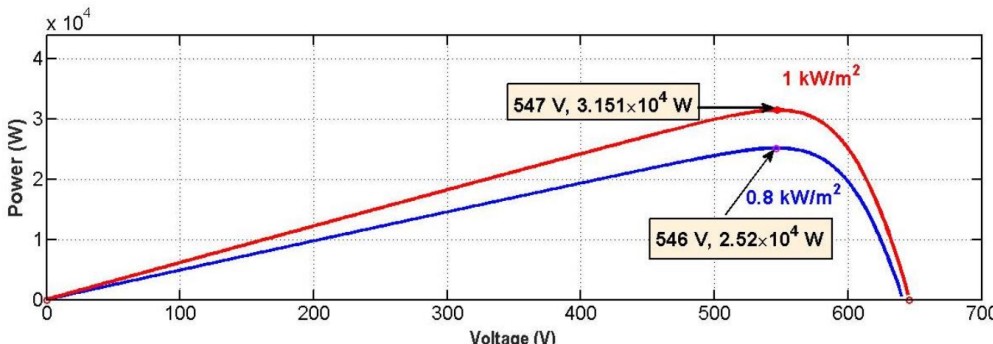

**Figure 14.** The P-V curve showing MPP at solar irradiance between 800 W/m$^2$–1000 W/m$^2$.

## 5. Conclusions

This paper set out to design a DERs standalone microgrid that feeds DC loads. It has discussed the modeling of the solar PV, WT-PMSG, and the DC/DC boost converters which couple these DERs to the DC microgrid. To ensure maximum power output from the solar PV and the WT-PMSG, MPPT algorithms were developed and implemented. This paper started out with the aim to regulate the DC microgrid voltage, thereby the MPPT algorithms were augmented by PI controllers so that the hybrid RES could achieve two things, maximum power tracking and voltage output error compensation via feedback control. The designed MPPT employed the P&O algorithm, while the PI controller gains were tuned by the Ziegler–Nichols method. This hybrid RES system with its controllers was implemented in MATLAB/Simulink simulation software. The simulations were undertaken for two scenarios of constant DC load and step changes of DC load. There were also varying irradiance, and varying wind speeds. Simulation results show that the output voltage of the DC microgrid tracks the reference fairly well.

This developed prototype could further be tested on a microgrid test bed. However, due to untenable constraints, the simulations alone are presented here. Future work will involve testing the whole DERs microgrid on the test bed and validate its performance. As a proof of concept, the developed microgrid control algorithm will be experimentally validated against other methods such as particle swarm optimization and genetic algorithms in the near future. Authors will also look at the efficiency aspect of the DC microgrid and investigate possible ways to optimize its efficiency. This is useful research for electrification of islanded areas far from grid connectivity, such as that DC microgrid at Luxmanda village.

**Author Contributions:** Conceptualization, M.I.J. and B.M.M.M.; methodology, M.I.J. and C.J.M.; software, M.I.J.; validation, M.I.J., B.M.M.M., C.J.M. and A.T.M.; formal analysis, M.I.J. and A.T.M.; writing—original draft preparation, M.I.J. and A.T.M.; writing—review and editing, A.T.M. and C.J.M.; supervision, C.J.M. and B.M.M.M.; project administration, C.J.M. and M.I.J. All authors have read and agreed to the published version of the manuscript.

**Funding:** This research was possible due to financial support of the Royal Society for PhD studentship, Registered Charity No. 207043.

**Institutional Review Board Statement:** Not applicable.

**Informed Consent Statement:** Not applicable.

**Data Availability Statement:** Data is contained within the article.

**Acknowledgments:** This research is supported by the Royal Society–DFID in the UK under the Africa Capacity Building Program Initiative. It is implemented by the ACERA consortium: that is the University of Leeds in the UK; CREEC-Makerere University in Uganda; Dar es Salaam Institute of Technology (DIT) in Tanzania and Marien Nguabi University in Congo Brazzaville.

**Conflicts of Interest:** The authors declare no conflict of interest.

## Appendix A

**Table A1.** Solar array sun power SPR-315-WHT-D.

| Parameters | Values | Units |
|---|---|---|
| Max. power | 315 | W |
| Current at MPP | 5.76 | A |
| Voltage at MPP | 54.7 | V |
| Temperature coefficient of $V_{oc}$ | −0.27269 | %/°C |
| Open circuit voltage $V_{oc}$ | 64.6 | V |
| Short circuit current $I_{sc}$ | 6.14 | A |
| Current/temp. coefficient, $K_i$ | 0.061745 | %/°C |

**Table A2.** Parameters of DC/DC converters.

| Parameters | Values | Units |
|---|---|---|
| Boost converter for solar PV | | |
| Inductor $L_{PV}$ | 20 | mH |
| Capacitor $C_d$ | 150 | μF |
| Resistive Load | 15 | Ωι |
| Boost converter for wind system | | |
| Inductor $L_w$ | 35 | mH |
| Buck–boost converter | | |
| Inductor $L_b$ | 1 | mH |

**Table A3.** Lead acid battery capacity.

| Parameters | Values | Units |
|---|---|---|
| Nominal Voltage, $V_b$ | 240 | V |
| Rated Capacity | 248 | Ah |

**Table A4.** Parameter for PMSG.

| Parameters | Values | Units |
|---|---|---|
| Rated speed | 153 | rad/s |
| Armature resistance, $R_s$ | 0.425 | Ω |
| Flux | 0.433 | Wb |
| Rated current | 12 | A |
| Stator inductance $L_s$ | 8.4 | mH |
| Rated torque | 40 | Nm |
| Rated power | 6 | kW |

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
