# Peer review of "Design of a Hybrid Energy System with Energy Storage for Standalone DC Microgrid Application"

_energies, doi:10.3390/en14185994_

Round 1

Reviewer 1 Report

The paper "Proposal Design of a Hybrid Solar PV-Wind-Battery Energy Storage for Standalone DC Microgrid Application",  proposes a microgrid distributed energy resources (DERs) for a rural standalone system. However, there are a few concerns that must be addressed before this article can be accepted.

  1. The title of the paper is not clear. Some like "Design of a Hybrid Energy System with Energy Storage for Standalone DC Microgrid Application." will  be more suitable. But in the current state the title is not only confusing but also not clear. Please modify to a suitable and correct title.
  2.  The Introduction section of this article is very short and doesn't cover the current research. Authors need to explain more about the current research, rather than only picking a few. Also, there is no mention of the contribution of the paper, short introduction of methodology, contribution and results are necessary. I suggest author's must review the academic writing style before submitting the article.
  3. The  Materials and Methods does provide the theoretical background, but it is not clear what is the contribution of this paper other than just simulation? 
  4. The originality of this paper is questionable, as the most of the simulation components are borrowed from other research and there is no section explaining how this article improve them or integrate them to create innovation?
  5. If the author's are doing a case study on Tanzania, they should at least use meteorological data from this region to validate the simulation that way this article will be relevant to the region. However, Section 4.1 and Section 4.2 mentions the simulation based on step wise solar and wind conditions. 
  6. The meteorological data are openly available at ERA5 dataset(https://www.ecmwf.int/en/forecasts/dataset/ecmwf-reanalysis-v5). Which makes the region specific simulations easier to perform. I will advise authors to reconsider the simulations using the region specific dataset to make the article scientifically sound and relevant to the region or target area.
  7. The conclusions doesn't discuss the novelty, contribution or summary of the results. 

Reviewer 2 Report

This paper topic is timely and important. The design of a DERs standalone microgrid that feeds DC loads has to be improved to accommodate various sudden circumstances under different conditions. The authors can consider more scenarios in winter and summer considering how fast their MPPT can deal with sudden change. Also, I think a separate part for efficiency analysis could help. This work can be done via simulation as well. The control algorithms used MPPT based PI controllers for maximum power tracking are not new.  Maybe gains were tuned by the Ziegler-Nichols method could be a good approach. So, I think adding one paragraph at the beginning to talk about the novelty of the work could help. The part of the power electronics converters rule is essential to be included in the paper. The references list has to be improved with more recent references. This developed prototype could further be tested on a microgrid test-bed. This is great if the authors can do it by at least compared with other experimental work. However, due to untenable constraints, the simulations alone are presented here. But, at least the authors can compare their work with other published experimental work to show the validity of their proposed design. Or, they can propose a complete comparison with other MPPT techniques starting from traditional ones like Perturb & Observe or IncCond, through utilizing PSO or Genetic or Artificial Intelligence techniques if possible. Please, add a complete section about future work and how the design could be improved in the future. These comments are proposed to improve the overall quality of the paper. Thank you for your good work. 

Reviewer 3 Report

Dear authors,

This paper presents a microgrid distributed energy resources (DERs) for a rural standalone system. It is made up of solar photovoltaic (solar PV) system, battery energy storage system (BESS), and wind turbine coupled to permanent magnet synchronous generator (WT-PMSG). 

The paper requires a double-check regarding its organization and language. 

Below are more comments to be considered:

1- The introduction should clearly introduce the problem statement and research gaps.

2- The contribution was not discussed in the introduction. Please add a part at the end of the introduction to discuss the main contributions.

3- The paper organization is not highlighted at the end of the introduction.

4- What is the novelty in the suggested MPPT?

5-Please compare with other methods where applicable.

Round 2

Reviewer 1 Report

Dear authors,

I would like to bring your attention to the comment for response for point 6

"The mentioned website https://www.ecmwf.int/en/forecasts/dataset/ecmwf-reanalysis-v5 has moved to a new data centre in Bologna as it says on the landing page. However, we have taken this and are currently collecting the meteorological data for Tanzania from the Tanzania Meteorological Agency and will proceed to validate the model. We hope to make this the second part as a continuation to this article, i.e., a new article with field data validation."

the ERA5 dataset on "https://cds.climate.copernicus.eu/cdsapp#!/dataset/reanalysis-era5-land?tab=overview" copernicus API is still functional. However as pointed out by authors that they have no intention of adding the Meteorological simulations in this paper. 

Thank you very much for clarifying the model is tested on "LINEAR" variation and not "STEPWISE" variation. However the use of capital letters was unnecessary and is highly discouraged in academic communication.

Reviewer 3 Report

accept